# Analysis and Comparison of Vector Space and Metric Space Representations in QSAR Modeling

**DOI:** 10.3390/molecules24091698

**Published:** 2019-04-30

**Authors:** Samina Kausar, Andre O. Falcao

**Affiliations:** 1LASIGE, Faculdade de Ciencias, Universidade de Lisboa, 1749-016 Lisboa, Portugal; saminakausar.bioinfo@gmail.com; 2BioISI—Biosystems & Integrative Sciences Institute, Faculdade de Ciencias, Universidade de Lisboa, 1749-016 Lisboa, Portugal

**Keywords:** QSAR modeling, non-contiguous atom matching structure similarity—NAMS, metric space, vector space, PCA, feature selection, random forest, support vector machines

## Abstract

The performance of quantitative structure–activity relationship (QSAR) models largely depends on the relevance of the selected molecular representation used as input data matrices. This work presents a thorough comparative analysis of two main categories of molecular representations (vector space and metric space) for fitting robust machine learning models in QSAR problems. For the assessment of these methods, seven different molecular representations that included RDKit descriptors, five different fingerprints types (MACCS, PubChem, FP2-based, Atom Pair, and ECFP4), and a graph matching approach (non-contiguous atom matching structure similarity; NAMS) in both vector space and metric space, were subjected to state-of-art machine learning methods that included different dimensionality reduction methods (feature selection and linear dimensionality reduction). Five distinct QSAR data sets were used for direct assessment and analysis. Results show that, in general, metric-space and vector-space representations are able to produce equivalent models, but there are significant differences between individual approaches. The NAMS-based similarity approach consistently outperformed most fingerprint representations in model quality, closely followed by Atom Pair fingerprints. To further verify these findings, the metric space-based models were fitted to the same data sets with the closest neighbors removed. These latter results further strengthened the above conclusions. The metric space graph-based approach appeared significantly superior to the other representations, albeit at a significant computational cost.

## 1. Introduction

In the past 50 years, quantitative structure–activity relationship (QSAR) has become a powerful tool for drug design and discovery. The underlying principle in QSAR modeling is the assumption that molecular structure information is sufficient to model and predict biological or pharmacological activity. Hence, in QSAR studies, different molecular representations have been used to describe the information encoded in molecular structures so as to predict the quantitative relationships between biological activity (response-variable) and structural information (predictors) [1,2,3,4,5].

The performance of QSAR models for the accurate characterization of biological molecular properties largely depends on the relevance of the selected molecular representation. Such representations can be divided into two broad categories of methods, namely, vector space and metric space representations [6]. A vector space or linear space representation occurs when the set of modeling instances is represented as a vector, with its characteristics measured relative to some reference frame and thus having a notion of magnitude and direction from the origin. In most QSAR modeling studies, vector space is the most common representation used, where each chemical structure is translated using a set of molecular descriptors. This is generally referred to as the “chemical feature space”, which represents different structural characteristics/properties [5,7,8]. Nevertheless, vector space-based QSAR modeling has two major modeling issues. The first is the determination of the set of features capable of structural representation and, the second is the identification of the subset of features that, more significantly, are able to predict the desired property [9,10,11,12,13]. Metric space representation, on the other hand, is built on the principle of measured distances between a set of instances that we want to model. As sometimes it is difficult to identify specific features of a real world entity such as a molecule, oftentimes it is easier to quantify its distance or similarity to other instances. A typical case for using metric space representations is in protein functional annotation; while it is quite hard to define a set of features that characterize a protein, the similarity between proteins (whether structural or sequence-based) is commonly used to assign its function, as it is known that above a given similarity threshold, proteins maintain their function [14,15]. In in silico screening, the similarity principle leads to the simplest database screening methods. If a seed molecule has been experimentally determined as active, the first approach to find other actives is to identify similar molecules, as the probability of finding other actives increases with proximity to the base molecule [16,17]. QSAR metric space modeling is also hampered by two different issues. In the first place, we need to determine how to measure similarity between molecules—for which there are currently several and conflicting approaches—and secondly, it is necessary to compute the distance of each molecule to all the molecules in the training sets, which may entail difficult computational problems. Distance matrices, as they are quadratic to the number of instances of the data set, add difficulties to the modeling effort and do not scale well, even with the increased computational power available today. Any vector space is a metric space, as it is possible to compute the distances between instances using any common distance metric such as the Euclidean distance. There are some data sets for which no vector representation is known (e.g., proteins); however, it is possible to compute their distance. Thus, all vector spaces are metric spaces, but the reverse is not true (Figure 1).

### 1.1. Molecular Similarity and Metric Space Representation

Molecular similarity largely depends upon an appropriate combination of two basic components including (a) a molecular structural representation to find the overlapping or similar features and (b) a similarity function/coefficient to quantify the similarity [18,19,20,21,22,23,24,25,26]. By far, the most commonly used structural representation for comparing molecules is the use of two-dimensional (2D) molecular fingerprints. Fingerprints are a sort of binary fragment descriptor, where each bit represents the hashing product of the possible chemical fragments of a molecule. There are currently several widely used fingerprints that differ in the form that a molecule is decomposed, the size of the representation, and the hashing algorithm [27]. Some other descriptor-independent methods are also available for molecular similarity comparisons, including molecular graph matching approaches [28,29,30,31]. To quantify molecular similarity, the most common method used is the Tanimoto (Jaccard) similarity coefficient [32,33]; however, there are many other similarity/distance methods [20,25,26,33,34]. The one-complement *D* of the Tanimoto/Jaccard coefficient, where D=1−J, has been proven to be a real metric, satisfying all the known properties of distance measures [35]. In comparison to vector space-based methods, there is limited research reported in the literature exploring the quantitative relationship between computed molecular similarity and activity in QSAR/QSPR modeling [7,16,19,36,37,38,39,40,41,42,43,44,45].

### 1.2. Metric Spaces vs. Vector Spaces

With all of the aforementioned concerns, the main question that we want to address in this study is whether a metric space or a vector space modeling approach outperforms the other in QSAR regression problems. Therefore, in this work, we have carried out a comparative analysis of molecular structural representation using some of the most commonly used vector and metric space-based methodologies and compare the results. Overall, we seek to answer the following four questions:Is metric space representation as good as the most common vector space-based approaches?Which similarity representation carries the maximum chemical/structural information content to establish the best relationship between structural similarity and activity?How effective is the reduction of dimensionality of the feature space with principal components by the replacement of explicit descriptors/fingerprints in QSAR modeling?Is there any one molecular structure representation method that is generally better than the others?

To accomplish these goals, the following work was performed: Five distinct data sets with distinct modelability characteristics were selected and curated from ChEMBL23. Several modeling efforts were then systematically applied to all selected data sets, namely (i) a typical vector space representation of molecules was performed by using an extensive set of chemical descriptors then used for model fitting in a QSAR optimization framework that includes automated data processing, descriptors/fingerprints computation, and feature selection; (ii) similarity matrices were computed for all data sets using a variety of methods (five fingerprint-based and one graph-based), and these similarity matrices were then used for modeling by using their principal components as model components; (iii) the fingerprint-based representations, as they actually also represent molecular features, were further used in a vector-based model, using the same linear dimensionality reduction method. For all three different modeling choices, the number of features (or principal components) used in each model was selected by using five-fold cross-validation, and each final model was assessed against an independent validation set randomly selected from the initial data set and which was never used in any step of the model-fitting phase.

## 2. Methodology

### 2.1. Overview of the Methodology

We collected and curated the molecular data for each biological target from ChEMBL23 [46], then all molecules of each data set were represented using different fingerprint models and molecular descriptors and separated into different modeling problems. To perform all of the analyses, each data set was initially randomly split into training and independent validation sets (IVSs), the former used for training and model selection, and the latter for the final evaluation of the model. A state-of-the-art QSAR modeling approach [47] was used to build a predictive model using an optimized feature selection procedure. The other models investigated with the same data sets required first the computation of five different fingerprint sets. These were used for additional vector space modeling and for the computation of similarity matrices between all molecules of each data set. Additionally, one graph-based structural similarity (NAMS) approach was used to make one further similarity matrix for metric-space modeling. Principal component analysis (PCA). was applied to both the similarity matrices and the bare fingerprints so as to create and evaluate models by iteratively increasing the number of principal components. The predictive performance of all data representations was assessed using the IVSs, which were never used during feature/PC selection (Figure 2). The details of each step of the followed methodology are covered in the following sections.

### 2.2. Vector Space Representation

In a vector space, each molecule is represented by using a feature vector that contains several molecular properties (descriptors) or structural features represented using a binary array of fixed size (fingerprints) [27,48].

#### 2.2.1. Descriptor-Based Representations

Molecular descriptors aim to selectively describe the information encoded in the structure [48]. Some molecular descriptors are derived with mathematical formulae obtained from chemical graph theory, information theory, and quantum mechanics, among other methods, that directly illustrate some relevant features of the molecules [48,49]. Molecular descriptors can be divided into four broader categories: constitutional (1D), topological (2D), geometrical (3D), and physico-chemical properties-based (4D) descriptors [48,50]. 2D descriptors are the most commonly used types of descriptors.

#### 2.2.2. Fingerprint-Based Representations

Another well-known molecular representation is molecular fingerprints, which are fixed-length bit-strings where each bit encodes a fragment or characteristic of a given molecule [27]. Molecular fingerprints are often very different in length and complexity, ranging from 2D/simple representations of relevant structural features to 3D/complicated pharmacophore arrangements. Thus, many types of fingerprints have been generated with different settings (generation method, length, size of patterns, and number of bits activated by each pattern, etc.) and are further deployed as descriptors for predictive modeling to estimate biological activities [12,27,51,52,53,54].

In principle, 3D representation should have higher information content than 2D, but surprisingly, higher complexity is often more error-prone and less robust in performance [26,55,56,57,58]. 2D fingerprints can encode different structural information, for example, molecular fragments and structural patterns, topological pathways through compounds, or topological atom environments either as bit strings or feature sets. Numerous software packages have been developed to generate several types of fingerprint for drug discovery applications [54]. Moreover, the basic principle of fingerprints generating algorithms and their comparative performance in a variety of QSAR problems has been extensively studied [8,26,54,59]. The preferred molecular fingerprints can be grouped into the following three classes:Topological/path-based fingerprints (e.g., Daylight-like RDkit [27,60] and Atom Pair [61]) capture the paths between atom types by describing their different combinations and always assign the same bit’s position to the same substructures within the compared molecules, which sometimes results in bit collisions but is also useful for clustering compounds.Circular fingerprints (e.g., ECFP [62]) record circular atom environments that grow radially from the central atom connections. In topological and circular fingerprints, an individual bit has no definite meaning.Structural keys fingerprints (e.g., MACCS [63], PubChem [64]), where each specific bit position represents the presence (1) or absence (0) of predefined functional groups, substructure motifs, or fragments.

2D fingerprints can easily be calculated by specialized, open-source, and readily available software packages (e.g., OpenBabel [65] or RDkit [60]). 2D fingerprint-based similarity analysis is the most widely used methodology in ligand-based virtual screening, clustering, and diversity analysis [24,26,59,66,67].

### 2.3. Metric Space Representation

A molecule in metric space is defined only as its relation (distance or similarity) to all other molecules in the data set. Technically, a metric space is computed using distances between all the elements of a data set, creating a distance matrix which can then be used in a variety of modeling techniques such as hierarchical agglomerative clustering or k-nearest neighbours models [68,69]. There is a variety of ways to transform similarities into distances [16,54]; however, as all the methodologies for comparing molecules produce similarity matrices, it was deemed unnecessary to transform the similarities into distances and we instead use similarity matrices directly for modeling, as this extra transformation would introduce one further step in the data preparation procedure with no clear advantage.

In descriptor-independent methods, graph matching approaches have been used. In these methods, graph theory is used to represent molecules as labeled graphs whose vertices correspond to the atoms and whose edges correspond to the covalent bonds. Several techniques, each with some advantages and limitations, are available to compare labeled graphs [29]. In the descriptor-independent methods, many advancements have been introduced to improve the sensitivity of graph matching methodology and obtain consistent and reliable molecular similarity results. One of these methods is non-contiguous atom matching structural similarity (NAMS), which has shown modeling advantages over other structural methods [28], although the computational cost of its application can be high.

#### 2.3.1. Fingerprint-Based Similarity

Many types of 2D and 3D molecular fingerprints have been generated to code chemical structures/properties into bit-string representations [20,70,71]. Molecular fingerprint representation allows for an easy comparison of molecules by identifying and quantifying the amount of overlapping elements between them. The applications of molecular fingerprints has been broadly reviewed and used in the literature [22,54,70,72]. There is a large variety of similarity and distance functions that have been introduced and return a molecular similarity score [54,59]. In cheminformatics, the prevalent approach is the use of the Tanimoto coefficient ($T_{c}$) over molecular fingerprints [26,33]. In the case of 2D fingerprint comparison, for binary vectors of fingerprints representing two molecules A and B, $T_{c}$ is defined as
(1)$$\begin{array}{c} Tc\left(A,B\right)=\frac{A\cap B}{A\cup B}=\frac{c}{a+b-c}. \end{array}$$

In Equation (Equation 1), *a* corresponds to the number of bits set to 1 in molecule *A*, *b* is the number of bits set to 1 in molecule *B*, while *c* is the number of common set bits in both molecules. $1-T_{c}$ is an actual distance measure, encompassing all four property distance measures referred to above.

#### 2.3.2. NAMS-Based Similarity

NAMS is a graph matching algorithm that uses a new atom alignment method to quantify the structural similarity between compared molecules [28]. NAMS breaks complex molecular structures into simpler parts to reduce molecules to atom–bond–atom structures and calculates a global structural similarity score from the best optimal alignment between the atoms of compared molecules. This algorithm has shown a higher discriminant power for biological activity than other structural or graph matching approaches. One of the reasons is that the applied atom matching methodology is able to consider important characteristics of atoms and bonds such as chirality and double bond stereo-isomerism that are oftentimes ignored in other approaches.

Given the structural representation of any two molecules, NAMS is able to compute its similarity score. NAMS can be fine-tuned with several parameters that allow users fo increase the importance of any specific molecular characteristics (atom or bond similarities and atomic characteristics like atom stereo-isomerism or double bond cis-trans isomerisms). Changing the parameters will change the resulting molecular similarities, but the overall results of comparing large and diverse data sets are not very much changed. For the current work, only the parameters were used.

### 2.4. Model Building

In QSAR modeling, the most well-known machine learning approaches include neural networks (ANN), support vector machines (SVM), decision trees, random forests (RF), and k-nearest neighbours [73,74]. In the last few years, RF [75] and SVM [76], two non-linear supervised learning methods, have become the most prevalent algorithms in QSAR studies [75,77,78,79,80,81,82]. One of the biggest advantages of SVM is its ability to deal with high dimensional and duplicated data with a lower risk of model overfitting [79,80,81,82], while, on the other hand, RF are considered specially robust in complex situations of high dimensional QSAR/QSPR data sets [75,77,78]. Hence, RF and SVM are the basic algorithms used in the learning phase of the current work.

As stated, some of the most prevalent issues in QSAR modeling approaches are variable redundancy or collinearity, with complex correlation patterns between descriptors or the presence of irrelevant features in the data set, which may reduce the quality of the produced models. These are consequences of the high dimensionality of such problems. Such issues are aggravated by the fact that in QSAR studies, there are oftentimes many more predictors than the number of actual instances to fit [9,10,11,12,13], which will make it more difficult to find adequately fitting models. Several approaches have been followed in the literature to solve the descriptor selection problem in QSAR modeling [73,77,83,84,85]. These approaches can be roughly divided into two different categories: feature reduction and feature selection. In feature reduction, the main purpose is to algebraically combine sets of features into statistically independent new components. There are several methods that purport to accomplish these goals, among which is principal component analysis (PCA) and singular value decomposition or kernel PCA [86]. PCA is by far the most commonly used method in feature reduction, while kernel based PCA is beginning to get some traction in the literature [87]. Feature selection, on the other hand, is a more complex problem, and in essence can be summarized as finding and selecting the smallest set of features that are capable of producing the best model. Methods to address this problem include the identification of linear correlations between all variables, bootstrapping methods capable of deciding which variables have the highest impact on model quality, or the use of optimization meta-heuristics like genetic algorithms [73,77,83,84,85].

In this work, we used two of the most common methods for feature reduction. PCA was used with the metric space data produced from the similarity matrices and fingerprint data, while random forests were used to identify the most relevant features capable of producing the highest-scoring models.

#### 2.4.1. Feature Reduction with PCA

Principal component analysis (PCA) is a linear reduction method used to calculate the most meaningful basis on which to re-express high dimensional data into a reduced space. However, PCA is a useful tool in QSAR modeling to deal with the problem of high data dimensionality and collinearity [4,68]. In typical QSAR studies, PCA is used to analyze the original data matrix in which molecules are represented by several types of predictor variables (molecular descriptors/fingerprints). PCA performs dimensionality reduction by transforming original descriptors’ space into linear orthogonal combinations of original variables named principal components (PCs). The generated PCs are uncorrelated and always ranked according to the decreasing data variance of the original variables [68]. As the first components contain the highest amount of data variance, models can be fit to data by gradually incrementing the components in the model. A first model will use only the first component, a second model will use the first two components, and so on, and which of these models with reduced dimensions is capable of producing the least amount of error in k-fold cross-validation is evaluated. Since each PC is an independent source of the original data variance, PCs have been used as a model input mainly when high data dimensionality is a big issue, and most models are sensitive to the number of variables used [68]. Several studies in the literature apply PCA for dimensionality reduction in QSPR/QSAR problems [4,88,89,90].

In this study, we performed PCA in both vector space representations (descriptor and fingerprint data matrices) and metric space representations (fingerprint-based similarity data matrices and NAMS-based similarity data metric). The generated PCs were used to build QSAR models with dimensionality reduction (DR). We compared the predictive performance of QSAR models generated by the reduced dimensionality of metric space with typical PCA-based QSAR models where vector space is reduced by PCA.

#### 2.4.2. Feature Selection with Random Forests

A random forest (RF) is an ensemble supervised nonlinear machine learning algorithm for classification or regression [75]. This algorithm generates a set of weakly independent decision trees that are built using randomly selected subsets of the data. Each generated tree is produced by randomly selecting a set of predictors from the full set and by sampling with replacement instances from the same data pool. This will create a set of randomly generated trees (a forest), each one created from different data and variable partitions. The RF algorithm then uses a consensus voting procedure to combine the predictions from all randomly generated weak models and make more robust predictions. One of the consequences of this bootstrap procedure is that it is possible to assess the power that each variable has in the final predictions. The trees that include such variables will typically have higher prediction power, and as such, it is possible to rank each variable in terms of its overall importance to the model quality. Many studies showed that RFs’ voting procedure can be used for feature selection by ranking and selecting each variable according to its importance in RF models [77,85,91]. In this ensemble method, each variable’s importance score is calculated using several variable importance (VI) measures. In regression problems, an increase in the mean squared error of a tree is one of the widely used VI measures, which explains how much prediction error increases with the random permutation of any given variable while keeping all others unchanged in a node of a tree [75,85,91,92].

In this work, we followed the random forest (RF)-based feature selection method [77] to rank features in a high dimensional vector space according to their importance score. These are then later used in the feedforward feature selection procedure (Figure 2).

#### 2.4.3. Support Vector Machine

An SVM [76] is a supervised machine learning algorithm that has been widely used for classification and regression-based data analysis in many fields, including QSAR studies [77,79,80,81,82]. For a given set of data instances, a discriminative SVM algorithm focuses on the identification of support vectors (data instances) to draw a decision hyperplane in a high dimensional space that best separates data instances with maximum margins. SVM uses different kernel functions for data transformation in a new hyperplane; these can be linear, radial basis functions, sigmoid, or polynomial, which are generally considered good choices for a majority of problems. The discovery of support vectors greatly depends on the selected kernel function. In contrast to other methodologies where there is a learning phase that heuristically searches thorough the multidimensional feature space, in SVM learning this search procedure is a mathematical optimization procedure, and it is guaranteed that an optimal solution can be found in polynomial time. This also implies that, as no random component is involved, the same solution model will be produced for each model. In this work, we used SVM in the process of feedforward feature selection where PCs from vector/metric reduced dimensionality space and RF importance score-based ranked variables from features/vector space were stepwise subjected to the SVM, and final QSAR models were developed with an optimized set of selected dimensions (Figure 2). In the current work, the radial basis function was selected for all problems.

#### 2.4.4. Model Evaluation and External Validation

N-fold cross-validation or model internal validation is the simplest approach, where the training data set is randomly divided into a number (*N*) of folds (parts), and each part is used as an external set for the validation of the predictive model, which was fitted by using the remaining compounds in the other N−1 partitions. Cross-validation is essential to optimize modeling parameters and variable selection, and to verify the internal predictive power and robustness of the QSAR model [89]. In our analysis, we performed N-fold cross-validation to find an optimized number of most relevant variables (variable/PCs selection). For this purpose, a feedforward approach was used to generate estimation models by sequentially adding the RF importance score-based ranked variables (more relevant to least significant) and PCs extracted from vector and metric spaces as an input in the SVM algorithm. The internal predictive performance of each model was assessed by computing the percentage of variance explained (PVE) and root mean squared error (RMSE) of each predictive model in cross-validation [93]. As the cross-validation may result in a different number of best-performing variables for different folds, an average of the PVE score was recorded across all folds each time. Finally, the set of dimensions that led to the smallest average predictive error score in all folds was considered as the selected number of descriptors/fingerprints/PCs. After performing all of this feature optimization, the whole training data set was reused to develop a model with the selected features to perform a blind external prediction using the independent validation set.

## 3. Data

We tested the proposed QSAR modeling methodology on five data sets for common human biological targets, retrieved from ChEMBL23 [46]. These were selected independently of any previous hypothesis (Table 1). We used an automated QSAR modeling workflow [47] to collect and curate data for each selected target. The bioactivity data of the selected targets was retrieved using the UniProt accession number (Table 1).

Moreover, missing data, salt groups, and mixtures (e.g., in unconnected molecules, smaller fragments were excluded) were removed. In duplicated data, if more than one record was present for the same compound, the one kept would be its most recent measurement, according to the publication year. All data sets feature Ki as the bioactivity measure. However, the logarithm of Ki is more typically used for modeling and makes more biological sense. Also, to encompass several problems of the more extreme values, it was decided to clamp the values between an interval so that very weak or possibly inactive molecules receive the same low score, while it is oftentimes unnecessary to discriminate results with Ki≤1 nM, as these are very active molecules. Thus, the following expression (Equation (Equation 2)) was used for all data sets to transform Ki into spKi (scaled and clamped pKi):(2)$$spKi=\left\{\begin{array}{cc} 0, & \mathrm{if}\;Ki\geq \;\;10,000\;\;\mathrm{nM},\\ \frac{4-\log_{10}\left(Ki\right)}{4}, & \mathrm{if}\;1\;\mathrm{nM}\;<Ki<\;10,000\;\;\mathrm{nM},\\ 1, & \mathrm{if}\;Ki\leq 1\;\;\mathrm{nM} \end{array}\right.$$

spKi values are thus clamped between 0 and 1, the most active compounds having values closer or equal to 1, and the lesser active or inactives will have values of zero. This clamping assumes that Ki values below 1 nM are considered extremely active compounds, while molecules with Ki values above 10,000 nM are considered very weak or inactive.

### Data Preparation for Vector and Metric Space Representations

For each data set, molecules were represented in metric and in vector spaces by using three different approaches: (a) common vector space methods using molecular descriptors or fingerprints, named vector space with FS (feature selection); (b) principal components over the similarity matrices, categorized as PCA on metric space; and (c) principal components over molecular descriptors and fingerprints placed in vector space, or DR (PCA) (Figure 2).

For vector space representation, we used 1348 descriptors (2D and 3D) calculated for each selected data set with the RDKit [60] toolkit (Appendix A). Separate modeling efforts were performed by testing five different types of fingerprints separately, including ECFP6 (circular), PubChem (substructure keys) computed using the CDK [94] toolkit, MACCS (substructure keys), RDkit (path-based), and Atom Pair (path-based) generated using RDKit [60]. The data preparation for principal component over metric space representation involved the computation of the similarity matrices between all elements of the training set and computing the distances of the IVS to those of the training set. Using the Tanimoto index, similarity matrices were obtained for each of the five fingerprints by adding the NAMS graph-based molecular matching algorithm. Models generated using dimensionality reduction of metric and vector spaces were named “optimized number of PC models” (OPC-models), as the procedure emphasizes selecting the best number of PCs, capable of producing more reliable models. Predictive models built using vector space with FS were named SF-models (model having the selected number of features) (Figure 2).

Thus, a total of eighteen different molecular representations were used in this study and served as input data to a machine learning algorithm for the generation of ninety regression models for five selected QSAR problems.

## 4. Results

### 4.1. Implementation of Analysis

All molecular descriptors and fingerprints used in this study were calculated using CDK [94] and RDKit [60] built-in nodes of the open-source data-mining framework KNIME (version 3.2) [95]. All analyses were performed using R (version 3.4.4) [96] on a desktop workstation powered by a 6th-generation Core i7 Processor (3.41 GHz) with 16 GB RAM. Package e1071 [97] was used for the SVM algorithm and an R library, randomForest [98], for RF. Both SVM and RF algorithms were implemented with the default parameters. The R package factoextra was used for dimension reduction using PCA [99]. It is noteworthy that in the PCA-based QSAR modeling, orthogonal projections/PCs for test sets in N-fold and IVS were calculated by using R’s PCA predict() function.

### 4.2. Results of Generated Models

OPC-models and SF-models were fitted with the training data sets of all selected targets. For all data sets, the training data was used to evaluate and select the model that was able to produce the smallest RMSE or PVE (ratio of the variance explained). Typically, this involved selecting models with a reduced number of features or PCs (Appendix A). The final models after feature selection were validated using the same IVS for each problem set (Appendix A).

The first aspect that stands out from these results is that the most relevant factor for explaining model quality is the nature of the data itself. The predictive performance of QSAR models highly depends upon different characteristics of the data set (e.g., size, chemical diversity, and presence of activity cliffs) [100,101,102,103,104,105,106]. As an example, the *HERG* data set can easily be seen as a difficult problem, independently of the approach followed to model it (Figure 3). On the other hand, the human Histidine Receptor 1 (*HRH1*) generally appears as more easily modelable, while the remaining three problems (*SIGMAR1*, *DRD3*, and *HTR2A*) show intermediate modelability characteristics. Secondly, with some relevant cases noted below, no single method uniformly performs better than the others, and each method’s performance seems to be heavily dependent upon the data set characteristics.

To have a more encompassing view of the produced results, we performed a Friedman ranked test [107]; this is a non-parametric test used to assess different treatments applied to different test situations, as is the current case. In the present situation, a modeling approach is considered a treatment, which is evaluated by its results for the different data sets. Each model is then ranked according to its performance, where the best models have a lower rank and vice versa. The Friedman test is then able to evaluate each performance according to its rank in all data sets, thus effectively providing a performance value for each modeling approach. Another advantage of the Friedman test is that it allows for a post hoc analysis that is able to better qualify the differences verified between treatments, for instance, by grouping similar models with similar performance values. For each modeling data set, the rank in PVE of each modeling approach was calculated in R’s agricolae package (Figure 4) [108]. The test results showed that there were significant differences between treatments, with a Chi-squared test of 38.44 with 17 degrees of freedom giving a *p*-value of 2.2 × 10^−3^, which strongly suggests that there are statistically significant differences between the different modeling approaches.

The post hoc analysis of the Friedman test allows groupings of statistically indistinct treatments under the same grouping [107]. A treatment can belong to several groups. Figure 4 indicates to which groups each model belongs. The significance level used was 0.05, meaning that the model groupings are correct with at least 95% confidence. It can clearly be seen only the models that belong to grouping *e*—the one with model rankings consistently lower (thus indicating higher quality modeling approaches)—are NAMS metric space PCA and Atom Pair fingerprints with classical feature selection. Moreover, it can be observed that the use of RDkit fingerprints and molecular descriptors, both with metric space representation and PCA dimensionality reduction, consistently appear in the highest positions (worst models).

We further dissected these individualized results according to the four major questions that were the main objectives of our analysis. These questions are addressed one by one in the following sections.

#### 4.2.1. Is Metric Space Representation as Good as the Most Common Vector Space-Based Approaches?

To answer this question, the results of all three different approaches (simple feature selection and PCA dimensionality reduction in both vector spaces and metric spaces) were analyzed. A comparison of OPC-models generated using PCA on metric and vector spaces and SF-models built using vector spaces with FS showed that the predictive performance of each QSAR model was influenced by the selected type of molecular structural representation (Figure 5), which was expected and consistent with the literature [50,103,109]. We performed a similar analysis using the Friedman test over the ranks of the median values of each data modeling approach from the explained variance (PVE) of the fitted models using each respective IVS (Figure 5).

Feature selection over vector spaces has proven to be the most globally reliably modeling approach and appears to be significantly better relative to the use of PCA on the same data. Metric space PCA appears as somewhere in between, closer to the feature selection approach. The Friedman test for this data yielded a Chi-squared value of 6.0, which corresponded to a *p*-value of 0.049, just below the 0.05 threshold. With such results, it is fair to conclude that the usage of metric space data may compromise the quality of the models produced when comparing results to traditional vector space feature selection models, yet it clearly outperforms vector space PCA-based approaches. It is nonetheless striking that the highest-ranking method in the overall assessment is NAMS, a metric space-based approach, which may allow us to suggest that the other methods of calculating molecular similarities may be responsible for this decreased performance and may not be as adequate to compute molecular distances.

#### 4.2.2. Which Similarity Representation Carries the Maximum Chemical/Structural Information Content to Establish the Best Relationship between Local Similarities and Activity?

To analyze which similarity representation contributed more significantly to reliable predictive modeling, the overall performance of OPC-models generated using six similarity data matrices (NAMS, ECFP6, RDkit, Atom Pair, MACCS, and PubChem-based similarities) was evaluated again using the Friedman test (Figure 6). The ranking of each metric space-based approach was assessed for each data set and the overall quality of each model quantified through the use of the Friedman test and respective post hoc analyses. For the present case, NAMS clearly emerges as the best approach, followed closely by Atom Pair and ECFP6 fingerprints, the former appearing in the same group as NAMS. The Chi-squared test for the metric space-based approaches ranked comparison was 15.2 with 5 degrees of freedom, which corresponds to a *p*-value of 9.5 × 10^−3^. Thus, test results again suggest that NAMS molecular similarity is able to more reliably capture important structural information, which eventually generates a better quantitative relationship between local similarities and compound activity.

#### 4.2.3. How Effective Is Using a Reduced Dimensionality of the Metric/Vector Space with Principal Components, Replacing Explicit Descriptors/Fingerprints, in QSAR Modeling?

This question can actually be answered by observing the previous results. It seems clear that when directly comparing PCA to direct feature selection (Figure 5), the latter produces markedly better results, which strongly suggests that the dimensionality reduction achieved with PCA is a poor proxy for a better structured search for the most relevant descriptors in a modeling problem. Nonetheless, using PCs from the similarity matrix allows us to capture the same information available from vector space modeling. These results also highlight the capability of fingerprints to produce high quality models without the need for other chemical descriptors. Furthermore, the fingerprint-generating method appears critical for producing the most reliable models. As is clear from the above results, Atom Pair and ECFP6 fingerprints appear as the best fingerprint-based similarity approaches, while the RDkit and PubChem fingerprints consistently lag behind all other models.

#### 4.2.4. Is There Any Solution That Is Globally Better on a Variety of Difficult Problems?

From the above results, it is clear that there is no one single best approach for dealing with complex QSAR problems. Although metric space-based NAMS and Atom Pair come out in first place most of the time, they are not consistent for all data sets. For instance, Atom Pair fingerprint representation performs poorly for the HTR2A model, while NAMS does not appear on top for the DRD3 data set. Similarly, as mentioned above, there does not appear to be any intrinsic advantage in changing from a fingerprint vector space-based approach to similarity-based metric space modeling. The most consistent result was that the use of PCA with descriptor data was generally a poor modeling approach. PCA can nonetheless be used with distance matrices to capture reliable information for modeling.

## 5. Discussion

Many studies have demonstrated that the selection of molecular structural representation has a larger impact on the predictability of QSAR models than the choice of model optimization methods [8,26,54,59,67,109,110]. Our results confirm these findings further, suggesting that reduced metric space representation using NAMS-based similarity and Atom Pair fingerprints with feature selection are the methods that more consistently address a variety of modeling problems.

Nonetheless, one further concern over such studies is how much novel information is actually being discovered from the models, as it is a known fact that similar molecules tend to have similar biological properties. Therefore, a distinct possibility is that the use of similarity matrices for inference may be result in reliable predictions only when molecules very similar to the training data set are present. Thus, one further test for these modeling approaches is to understand how reliable these methods are for making models where all very similar molecules have been removed and no molecule, either in the training set or the IVS, has a high similarity to any other. This would allow the evaluation of the capability of each approach to make inferences when very diverse compounds are fed into the model. Therefore, to check the robustness of the tested methodologies, the five data sets were manipulated by converting them into harder problems with only structurally diverse molecules, making certain that no molecules within a given similarity threshold are present in each data set. Accordingly, five new data sets were created based on the initial ones but where no molecule was present if it was similar, within a given threshold, to others already present. As different similarity methods produce different scores for the same molecules, the thresholds were adjusted for each similarity method to make sure that the model would be trained with a similar number of instances (Table 2). This complementary analysis obviously relates only to metric space modeling, thus the following results will only focus on this modeling approach.

After removing the nearest neighbors, all data sets were again randomly split into training and independent validation sets, and the same data processing procedures were repeated for these new, more challenging data sets. Moreover, the same modeling principles were repeated by training the models with the training set while simultaneously selecting the best feature set and finally validating the best model with the corresponding IVS. The overall performance of the same models using these new data sets was assessed. Because the number of instances present in all new problems is different, both RMSE and PVE were used to adequately assess each model’s performance (Figure 7).

As can be seen, with such hampered data the performance of QSAR models has naturally dropped, leading to a decrease in PVE ranging from 0.15 to 0.52 (Figure 7A). This finding is consistent with the literature [8] in that similar molecules present in models tend to inflate result statistics. It can also now promptly be seen that the differences between the different models are now amplified, and it is clearly easier to visually identify which approaches distinguish themselves from all others. Nonetheless, the overall model ranking was not significantly changed. Thus, NAMS similarity representation was, for these data sets, clearly the highest-performing model, achieving the lowest RMSE scores in all cases. Using the Friedman interquartile range graph (Figure 7B) performance scores for both Atom Pair and ECPF6 are dependent on the use of PVE or RMSE. All other fingerprint approaches were not up to the referred methods used in these more difficult challenges. The Friedman test for the PVE had a Chi-squared value of 21.1, with a *p*-value of 7.8 × 10^−10^.

### Computation Time

The execution time of QSAR models built from reduced dimensionality of metric space ranged between 60.61 and 48.88 min and for vector space, 52.53 to 15.34 min, whereas vector space with FS computational time ranged between 860 min (DRD3) and 17 min (Sigma1R). A comparative analysis of computational time showed that reduced dimensionality significantly reduced the complexity of the problem at hand and that computational time cost also decreased.

Computation time is an important issue when comparing different modeling approaches, especially when the use of metric space methods is being evaluated, as the use of a full similarity matrix is required for each data set. Furthermore, metric space modeling requires that one of the steps for inference is that the distance of each new molecule to all of the molecules in the training set is assessed. This is not typically a problem for academic studies but may put a large computational burden for actual screening efforts when several millions of molecules are being evaluated. This problem is aggravated in the case of the specific non-fingerprint approach we tested (NAMS). Although apparently able to produce a more accurate distance, which translates into better prediction models overall, it does so at a much higher computational cost. With current common hardware, the average computational cost to compute the similarity of two molecules is 12 ms, which for many problems may be too high for many problems. As an example, computing the similarity of one new molecule to a training set of 1000 molecules will require 12 s. Such computational costs (although the problem is trivially parallelizable) may involve unacceptable computational costs for very large data sets.

## 6. Conclusions

In this study, we compared different molecular representation approaches for input into QSAR machine learning methods. These approaches were divided into vector space- and metric space-based, with each molecule being represented as a vector of different characteristics in the former, and with a molecule being represented by its distance or similarity to others of known activity in the latter. We have tested five different fingerprint types (RDKit-FP2-based, MACCS, PubChem, Atom Pair, and Morgan’s ECFP6) both as vectors of descriptors and, in metric space approaches, with Tanimoto scores computed for similarity. One exclusively vector space approach was also tested, where common chemical descriptors were computed and used in vector space modeling, as well as a pure metric space method with a molecular graph-based similarity (NAMS). We also tested whether it was more adequate to use dimensionality reduction methods (as with PCA) or a more computer-intensive feature selection procedure. These representation and dimensionality reduction methods were tested over five different data sets of different modelabilities and analyzed by the Friedman test for ranking models. Results showed that the choice of molecular representation to compute molecular similarity is more important than the modeling approach followed, thus certain methods produced consistently better results. ECFP6 and Atom Pair fingerprints were clearly the best approaches for modeling in vector spaces, surpassing all other methods. Classic molecular descriptors did not show any advantage for any of the data sets in this study. Regarding dimensionality reduction methods, the use of principal components appeared to be inferior to the use of random forest-based feature selection. The former method, albeit faster to process, generally produced results not on par with the latter.

In this study, metric space modeling by itself did not appear clearly superior to a vector space-based approach and, for the same representation, using fingerprints as descriptors tended to produce better results than using molecular distances from those same fingerprints. However, when using metric space representations, the differences between similarity methods become even more clear, with NAMS and Atom Pair fingerprints appearing objectively better than all other representations. When verifying whether metric space-based representations can be used for more remote inferences where the chemical space is evaluated in regions distant from the training data, the above conclusions regarding metric space modeling appear to have been amplified, and a larger distance between similarity methods was observed, with NAMS and Atom Pair fingerprints appearing clearly separated from the others.

Finally, metric space-based methods are more computationally expensive, requiring the computation of molecular similarity to every instance of the training set for each new molecule. This is a particularly severe cost for the graph-based similarity algorithm used (NAMS), and computation cost is a serious factor that may hamper its applicability in a real world virtual screening approach, despite overall being the method that is more consistently capable of producing high-quality QSAR models.

## Figures and Tables

**Figure 1 molecules-24-01698-f001:**
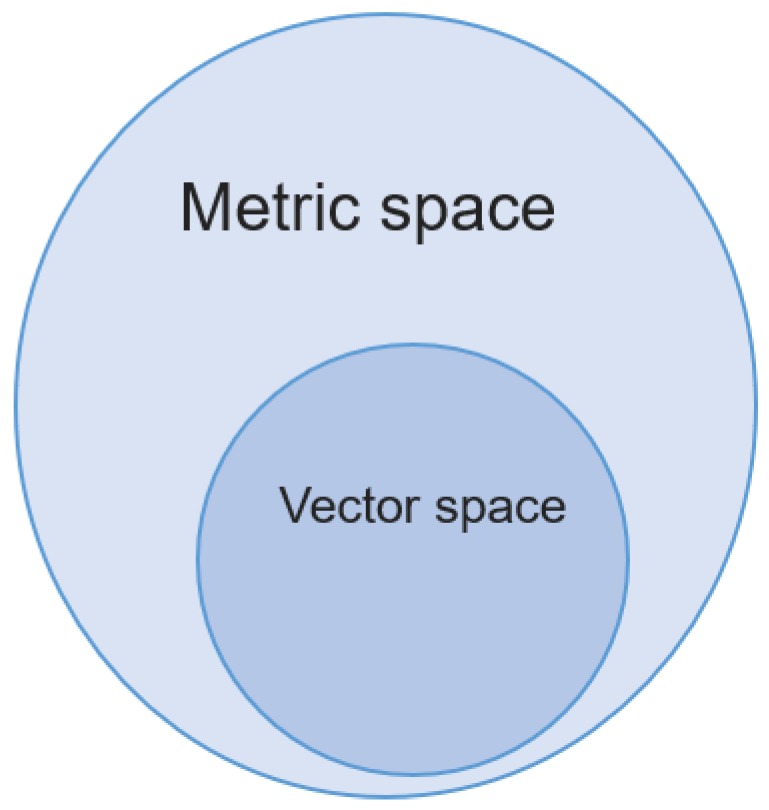
Vector space vs. metric space.

**Figure 2 molecules-24-01698-f002:**
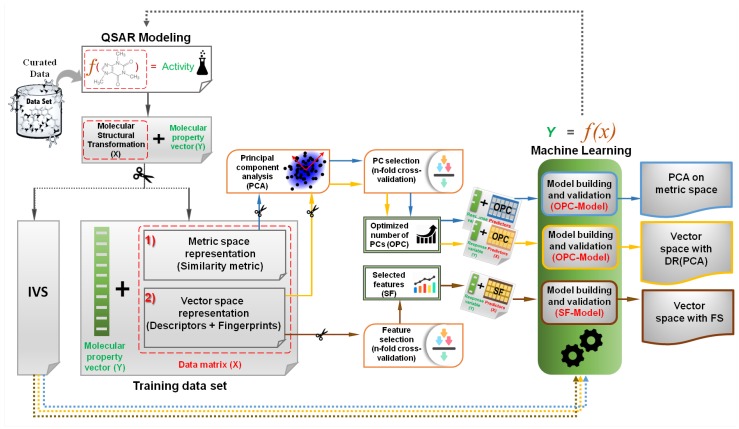
Quantitative structure–activity relationship (QSAR) modeling methods.

**Figure 3 molecules-24-01698-f003:**
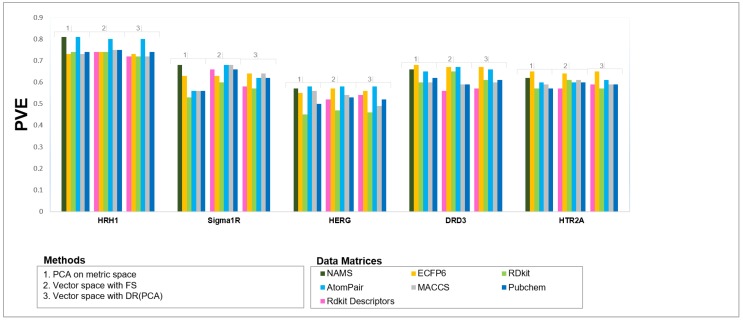
Comparisons of QSAR models’ predictive performance using independent validation sets (IVSs). PVE: percentage of variance explained by the model.

**Figure 4 molecules-24-01698-f004:**
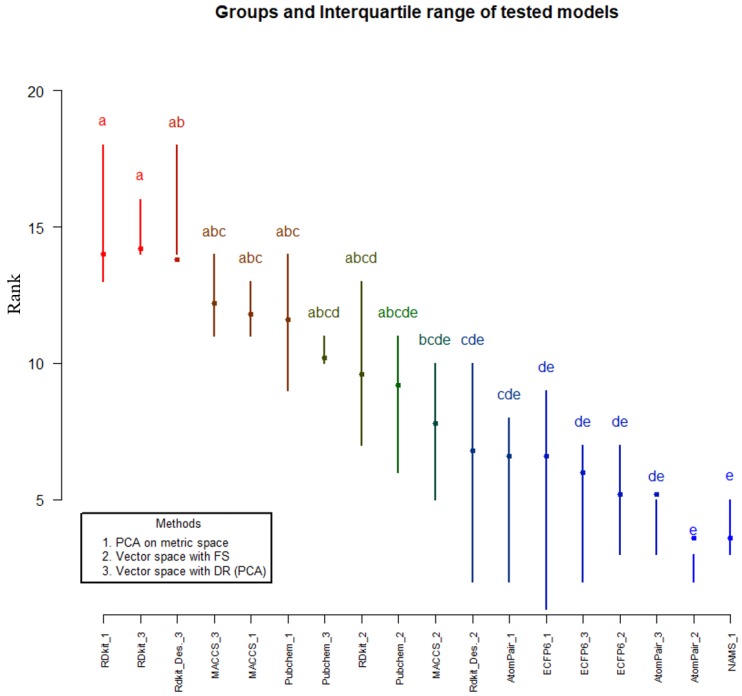
Friedman test results and interquartile ranges of tested models.

**Figure 5 molecules-24-01698-f005:**
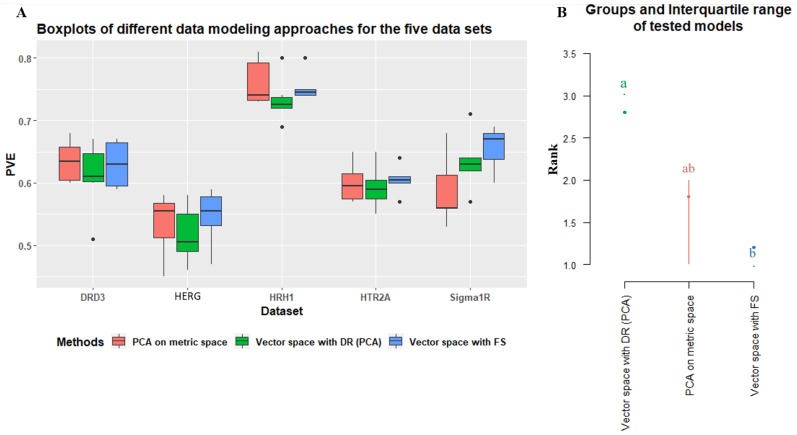
(**A**) Boxplots of the three modeling approaches grouped by the different data sets; (**B**) groups and interquartile ranges of the medians of tested models from the Friedman test post hoc analysis.

**Figure 6 molecules-24-01698-f006:**
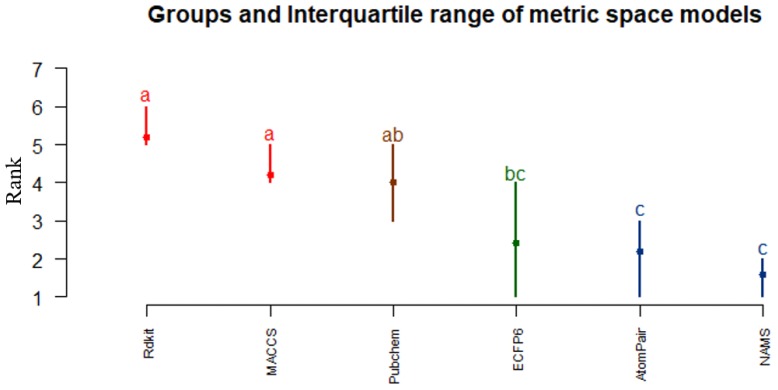
Overall performance of similarity representation using PCA on metric space-based QSAR modeling approach.

**Figure 7 molecules-24-01698-f007:**
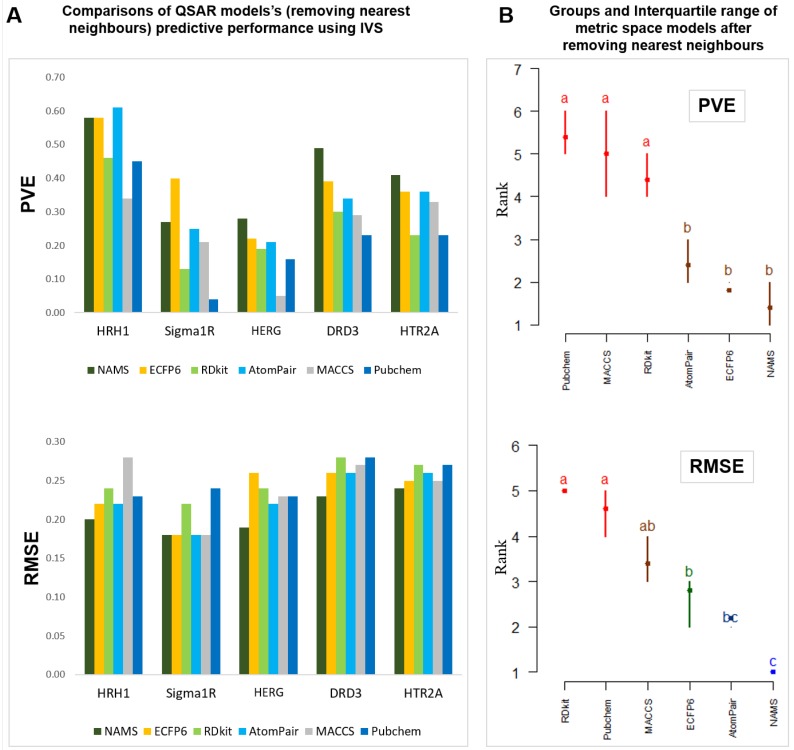
Overall performance of metric space representation after removing nearest neighbors in a PCA on metric space-based QSAR modeling approach.

**Table 1 molecules-24-01698-t001:** Data set description.

Uniprot ID.	Gene Name	Target Protein Name	Associated Bioactivities (Y)	Total Number of Observations (N-Processed)
P35367	HRH1	Histamine H1 receptor	Ki	1222
Q99720	SIGMAR1	Sigma non-opioid intracellular receptor 1	Ki	226
Q12809	HERG	Potassium voltage-gated channel subfamily H member 2	Ki	1481
P35462	DRD3	D(3) dopamine receptor	Ki	2902
P28223	HTR2A	5-hydroxytryptamine receptor 2A	Ki	2088

**Table 2 molecules-24-01698-t002:** Data size before and after removing nearest neighbors. Thr—similarity threshold; N—new data set size.

Target Protein Name	Data Size without Removing Nearest Neighbors	NAMS		ECFP6		RDkit		Atom Pair		MACCS		Pubchem	
		Thr	N	Thr	N	Thr	N	Thr	N	Thr	N	Thr	N
Histamine H1 receptor (HRH1)	1222	0.80	379	0.55	378	0.80	371	0.67	376	0.84	379	0.87	391
Sigma non-opioid intracellular receptor 1 (Sigma1R)	226	0.87	312	0.61	310	0.89	305	0.75	309	0.92	311	0.94	321
Potassium voltage-gated channel subfamily H member 2 (HERG)	1481	0.80	397	0.54	394	0.82	392	0.69	395	0.83	395	0.86	403
D(3) dopamine receptor (DRD3)	2902	0.80	478	0.52	481	0.77	470	0.67	480	0.87	484	0.86	484
5-hydroxytryptamine receptor 2A (HTR2A)	2088	0.80	432	0.47	432	0.78	424	0.63	426	0.83	429	0.85	437

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
