# Peer review of "Analysis and Comparison of Vector Space and Metric Space Representations in QSAR Modeling"

_molecules, 2019, doi:10.3390/molecules24091698_

Round 1
Reviewer 1 Report
The manuscript needs professional language editing before being considered for publication. The main concerns and recommendations are included in the attachment.

Author Response
Response to Reviewer 1 Comments
Answer: We gratefully acknowledge this reviewer efforts in such an extensive review of the manuscript and useful suggestions to improve our work. Many of the comments related to simple editing of the text have been addressed directly the revised manuscript and all the changes are highlighted in a supplementary file “Manuscript with highlighted changes.pdf”. Some of these comments which requires some justification or explanation were addressed individually:
Comment [4]: Are these terms coined by the authors or someone did introduce these before (In such case please add reference(s)).
Answer [4]: The term metric space and vector space are pretty well known in algebra. We did not feel the need to reference it as it is standard nomenclature (please check the Wikipedia entry) The classic text is
Victor Bryant, Metric Spaces: Iteration and Application, Cambridge University Press, 1985,
but for computational aspects the standard introduction to the problem is probably
E Chávez, G Navarro, R Baeza-Yate (2001) Searching in metric spaces. ACM computing surveys
where the problem is widely discussed albeit with a different emphasis (information retrieval). Nonetheless we have appended this reference to the text (“Manuscript with highlighted changes.pdf” line 6)
Comment [7]: I think that this term (i.e. Chemical Feature Space) is much better term than Vector Space. Chemical Feature space is intuitively self-explaining term.
Answer [7]: Indeed, it may be, but the vector space term is more general and more formally correct than “Chemical Feature Space”. Furthermore, our goal is to compare different vector based data modeling approaches with methods based on similarities (thus metric based). We thus used a standard naming that may be able to be readily understood by readers of different areas.
Comment [9]: This reviewer thinks that the aforementioned modeling issues for vector based QSAR are still valid for metric space representation. In the following text authors defending this hypothesis – please make the case stronger. For example: if the set of features will be not representative for the modeling of given property the similarity feature set will not help much in modeling.
Answer [9]: Not necessarily. Metric space modeling will add challenges that are not present in vector space based approaches. For instance, Feature selection is now a fairly standard procedure in QSAR studies, which is only possible in Vector Space data. Also, the fact that the data matrices grow quadratically with the number of instances (molecules) will entail modeling challenges for metric space data. Yet in some situations metric approaches may be the best solution with no corresponding vector space alternative. As our study hopefully showed, NAMS is an exclusively metric space approach, yet is able in most cases to outperform other approaches. We hope that the new revised version of the manuscript, with the methods section following the introduction will make this point self evident.
Comment [13]: These days it should not be a problem especially when we have relatively easy access to cloud, cluster and parallel computing force. In discussion authors say: ‘The execution time of QSAR models built from reduced dimensionality of metric space ranged between 60.61 to 48.88 minutes and for vector space 52.53 to 15.34 minutes’ – looks like not terrible difference (for getting great models).
Answer [13]: This point is addressed in the text which was modified accordingly (“Manuscript with highlighted changes.pdf” line 53 to 59): “Distance matrices as they are quadratic to the number of instances of the data set add difficulties to the modeling effort and do not scale well, even with increased computational power available today. Any vector space is a metric space, as it is possible to compute the distances between instances using any common distance metric as the Euclidean distance. On the other hand, there are some data sets for which no vector representation is known (e.g. proteins), while on the other hand, it is possible to compute their distance. Thus, all vector spaces are metric spaces, but the reverse is not true.”
Comment [17]: What do you mean by ‘1-complement …’? Is it the complement of T/J index found by subtracting T/J index from 100%?
Answer [17]: We clarified this in the revised version of the article (“Manuscript with highlighted changes.pdf” line 74 to 76). The one-complement D of the Tanimoto/Jaccard coefficient, where D = 1- J, has been proven to be a real metric, satisfying all the known properties of distance measures
Comment [31]: The number of optimized components In Table S1 is very often high (i.e. from 28 to 234 - depend on the dataset). Could authors include (in supplementary) one of the Variance/Scree (RMSE vs PCs#) plots from PCA analysis.
Answer [31]: We generated a new supplementary “Figure S1” that shows PVE vs number of PCs plot from PCA on metric space (“Manuscript with highlighted changes.pdf” line 381 and 604).
Comment [33]: hERG
Answer [33]: The suggestion was to replace HERG to hERG. In the manuscript, we are using genes name that's why “HERG” is italic within the text.
Comment [35]: What does it mean? No method is significantly better than others in all 5 sets of data? I can see that in most cases the modeling performance for the NAMS, AtomPair and ECFP6 is noticeable better. Please discuss it.
Answer [35]: What we wrote in the manuscript was that “no single method appears uniformly above the others” Which we meant that it is not self evident that a method overperforms all others (unless a close inspection of the graph is done). This is why we performed the Friedman test that was able to give us a quantitative measure of how each model ranked compared to others.
Comment [42]: What is ‘the discriminating alpha’ term?
Answer [42]: It was meant to be the “significance level” of hypothesis testing. We have corrected this within the text “The significance level used was 0.05, meaning that the model groupings are correct with at least 95% confidence.” (“Manuscript with highlighted changes.pdf” line 409)
Comment [46]: What is plotted on a Y-axis? What is the meaning of labels on top of each line/bar (i.e. a, ab, …, de, e)?
Answer [46]: All figures (Figure 4, 5, 6 and 7) showing Friedman’s test results have been updated with the Y-axis labels. The labels on top of each line/bar (i.e. a, ab, …, de, e) represent different groupings of models, for instance if different models appear within grouping a, it means that for the present analysis they are not statistically distinct (overlapping confidence intervals).
Comment [48]: Why fitted models and not performance measured for IVS?
Answer [48]: Model performance was always assessed using the corresponding IVS. We clarified this point in the manuscript (“Manuscript with highlighted changes.pdf” line 428).
Comment [49]: How the significance was estimated?
Answer [49]: Significance of each modeling approach was estimated through Friedman test that sums the ranks in each group and if sums are very different, the P value will be small (significant difference between groups).
Comment [50]: Please use the same color scheme (as is in A) for the PCA on metric space (red) and Vector space with DR (green).
Answer [50]: “Figure 4” in the revised version is “Figure 5” which has modified accordingly with the same colour scheme.
Comment [51]: On the basis of which premises is this conclusion?
Answer [51]: We are just making a suggestion; it is in fact not a formal conclusion. The fact that the NAMS Metric space appears clearly better than all other metric space approaches (see Figure F) may mean that the problem is not the methodology itself (use of metric spaces vs vector spaces) but rather the way in which the molecular distances are computed, which when using fingerprints may fail to capture some aspects critical for modeling.
Comment [58]: Threshold values in Table 1 range from 0.47 to 0.94. How the threshold values were calculated? Is it done by any optimization.
Answer [58]:Table 1 in the updated article is Table 2. This point is explained in the article. “As different similarity methods produce different scores for the same molecules, the thresholds were adjusted for each similarity methods, to make sure that model would be trained with a similar number of instances.” So, the similarity threshold was adjusted so that, for each similarity methodology, a similar number of compounds were selected for each data set
Comment [66]: PLEASE REWRITE the METHODOLOGY SECTION Most of the text presented in this section should be removed or moved to the other part of the manuscript. Please take into consideration that in the future, some of the readers may like to repeat or expand this study. Please convert this section to simple (but with enough details) recipe. Please do not focus on the background behind the origin of descriptors or methods used (in general) by other scientist. ‘Go to the point!’ According to the template document information included in ‘Methodology’ or ‘Materials and Methods’ should describe with sufficient details to allow others to replicate and build on published results. Please note that publication of your manuscript implicates that you must make all materials, data, computer code, and protocols associated with the publication available to readers. Please disclose at the submission stage any restrictions on the availability of materials or information. New methods and protocols should be described in detail while well-established.
Answer [66]: We agree with the reviewer that most of the part of the methodology is the background of the approaches. This was perhaps confusing because all the background was explained after the results section. Therefore, we rearranged the whole structure of the manuscript. In the methodology section, we are not covering all available methods of QSAR modeling like a review article. As the presented work is a comparative analysis of different molecular representation approaches in QSAR modeling, it is really important to adequately cover the background of the selected or compared methodologies. We hope the revised version will make this aspect more clear.
Reviewer 2 Report
The authors expose a quite complete and extensive m minireview concerning molecular descriptions and the consecution of QSAR models. They try to distinguish between metric and vector spaces, and also among the consecution of several models for several molecular sets. The work has been intense and complete regarding the predefined goals. The descriptions and reasoning are clear, and many of them come from general (and interesting) questions posed. The authors also consider the problem of model robustness. The references are complete and quite exhaustive.
Author Response
Answer to Reviewer 2 Comment
Comment 1: The authors expose a quite complete and extensive m minireview concerning molecular descriptions and the consecution of QSAR models. They try to distinguish between metric and vector spaces, and also among the consecution of several models for several molecular sets. The work has been intense and complete regarding the predefined goals. The descriptions and reasoning are clear, and many of them come from general (and interesting) questions posed. The authors also consider the problem of model robustness. The references are complete and quite exhaustive.
Answer 1:
We thank the reviewer for these comments. We believe that the overall goals of this paper were perceived and appreciated.

Reviewer 3 Report
If there is any novelty in this work, then it must be the use of the scissor icon cutting through the connectors of the absolutely obscure Figure 7. Not to mention "Molecular Structural Transformation" == that means "chemical reaction" in plain English. Frankly, I do not understand the difference between "metric spaces" and "vector spaces" - you can always calculate a metric in a vector space, and reversely you might, if you want, take the "pure kernel" methods, such as graph matching, and calculate some latent descriptors by embedding into some N-dimensional space, even if the initial similarity scores did not employ descriptors (vectors). You may, if you wish, apply explicit descriptor selection to similarity methods (say: this is my vector fo descriptors/fingerprints, but I decide to use only terms #6, 28, 29,49 and 67 to calculate the Tanimoto - so no, feature selection is NOT specific to vector spaces. All this makes no sense - except for the trivial conclusions that some sets are easier to model than others [where the more modelable are often easy to model because of intrinsic bias - for example, all actives are benzodiazepines while inactives are highly diverse!]. And yes - what really matters is having relevant structural information captured by the descriptors: an obvious truth. Obvious it is - but not because of this work, which sets out to ellucidate key questions of QSAR... based on N=5 study cases: 4 GPCRs and a ion channel! This is like studying four alcohols and an aldehyde and then... writing an organic chemistry textbook! QSAR is much larger a field - what about ADME/Tox, physico-chemical properties, etc? There is no "absolute" best descriptor space, and state-of-art machine learning delivers comparable results. This is something people in the field already know very well - because they saw it happening in HUNDREDS of cases, not five!
Author Response
Answers to Reviewer 3 Comments
Comment 1: If there is any novelty in this work, then it must be the use of the scissor icon cutting through the connectors of the absolutely obscure Figure 7. Not to mention "Molecular Structural Transformation" == that means "chemical reaction" in plain English. Frankly, I do not understand the difference between "metric spaces" and "vector spaces" - you can always calculate a metric in a vector space, and reversely you might, if you want, take the "pure kernel" methods, such as graph matching, and calculate some latent descriptors by embedding into some N-dimensional space, even if the initial similarity scores did not employ descriptors (vectors). You may, if you wish, apply explicit descriptor selection to similarity methods (say: this is my vector fo descriptors/fingerprints, but I decide to use only terms #6, 28, 29,49 and 67 to calculate the Tanimoto - so no, feature selection is NOT specific to vector spaces. All this makes no sense - except for the trivial conclusions that some sets are easier to model than others [where the more modelable are often easy to model because of intrinsic bias - for example, all actives are benzodiazepines while inactives are highly diverse!]. And yes - what really matters is having relevant structural information captured by the descriptors: an obvious truth. Obvious it is - but not because of this work, which sets out to ellucidate key questions of QSAR... based on N=5 study cases: 4 GPCRs and a ion channel! This is like studying four alcohols and an aldehyde and then... writing an organic chemistry textbook! QSAR is much larger a field - what about ADME/Tox, physico-chemical properties, etc? There is no "absolute" best descriptor space, and state-of-art machine learning delivers comparable results. This is something people in the field already know very well - because they saw it happening in HUNDREDS of cases, not five!
Answer 1: In the first place, we want to thank the reviewer for his/her comments. Our initial version suffered from several shortcomings which we hope to have corrected in this new extensively revised version, and may have hampered his/her perception of our work. Firstly, and most importantly, the “methods” section now follows the Introduction, which will make the full manuscript more readily understood in its purpose and goals. Furthermore, the text was fully revised and we believe that the new manuscript more adequately presents our goals and conclusions
Nonetheless, we will address in detail several of the comments made by the reviewer in the following paragraphs
Reviewer 3 comment: “If there is any novelty in this work, then it must be the use of the scissor icon cutting through the connectors of the absolutely obscure Figure 7”
Answer:
We appreciate the reviewer’s light hearted comment on the use of the scissors as a graphic metaphor for the data editing process, nonetheless, we emphatically disagree that there is nothing else novel in the manuscript, as we will address every issue raised.
Reviewer 3 comment: “Not to mention "Molecular Structural Transformation" == that means "chemical reaction" in plain English”
Answer:
That would be correct if you are talking about actual molecules, which is obviously not the case. We are discussing computational transformations of molecular representations. In the same way a “black hole” does not exactly mean a “black hole” in plain English if we are reading a physics text.
Reviewer 3 comment: “Frankly, I do not understand the difference between "metric spaces" and "vector spaces" - you can always calculate a metric in a vector space, and reversely you might, if you want, take the "pure kernel" methods, such as graph matching, and calculate some latent descriptors by embedding into some N-dimensional space, even if the initial similarity scores did not employ descriptors (vectors).”
Answer:
That is only half right and we actually point out the relation between metric and vector spaces in the manuscript. A vector space is in fact a metric space, but the reverse is not true. The example given in the manuscript of sequence alignment in proteins is a classic example. BLAST (or Smith and Waterman for that matter) is able to measure protein distances, however, there is no direct way to compute “descriptors” out of distances. Certainly, you can use heuristics, like kernel methods, Principal Coordinate Analysis - and we’ve actually used that in this paper - or one of the other myriad methods for multidimensional scaling, but we must always be mindful that in each transformation we are losing information. How much information? This is one of the findings in this paper - we were able to measure the impact of using a metric space data representation vs a vector space data representation as its effect in predicting pharmacological activity. This is one point that we believe is not handled anywhere in QSAR literature, and we hope to have contributed to elucidate this issue with this manuscript. We further hope that the new version of the manuscript is able to make this point absolutely clear, as the methods section further expands the essential concepts enumerated in the Introduction
Reviewer 3 comment: “You may, if you wish, apply explicit descriptor selection to similarity methods (say: this is my vector fo descriptors/fingerprints, but I decide to use only terms #6, 28, 29,49 and 67 to calculate the Tanimoto - so no, feature selection is NOT specific to vector spaces”
Answer:
That is totally besides the goal of this manuscript. In the fingerprint based approaches, the full Tanimoto distances were used. We also did not claim tha feature selection was specific to vector spaces. In the “vectorization” of the similarity matrices we used sequentially its linear principal components in a dimensionality reduction process that relates to the simple feature selection process used on the vector space data. We further compared the results obtained by using the same principal component approach with the vector data so that any bias resulting from this method could be isolated. This issue we believe was never so clearly addressed in a QSAR study and we believe that our results shed a fresh perspective over this topic. Again we hope that the fact of the Methods section now preceding the Results will make this aspect wholly clear
Reviewer 3 comment: “All this makes no sense - except for the trivial conclusions that some sets are easier to model than others [where the more modelable are often easy to model because of intrinsic bias - for example, all actives are benzodiazepines while inactives are highly diverse!] And yes - what really matters is having relevant structural information captured by the descriptors: an obvious truth.”
Answer:
Although we did not delve into benzodiazepines, that is actually a fundamental question that we addressed directly, differently from most published cheminformatics studies (and our approach is definitely new). To understand the relevance of very similar clusters of molecules, composed mostly of actives (or inactives for that matter), we did that analysis in the Discussion, where we presented the modeling results of the metric space representations in derived data sets where we removed all highly similar molecules to verify whether our conclusions would be maintained in such a much more difficult situation. Although the results were predictably worse, the overall conclusions of this study were maintained.
Reviewer 3 comment: “Obvious it is - but not because of this work, which sets out to ellucidate key questions of QSAR... based on N=5 study cases: 4 GPCRs and an ion channel! This is like studying four alcohols and an aldehyde and then... writing an organic chemistry textbook! QSAR is much larger a field - what about ADME/Tox, physico-chemical properties, etc? There is no "absolute" best descriptor space, and state-of-art machine learning delivers comparable results. This is something people in the field already know very well - because they saw it happening in HUNDREDS of cases, not five!”
Answer:
The goals of this study were not to write a cheminformatics textbook, not even a QSAR textbook. The major goal of this paper was only to address the essential question of whether or not there were advantages in metric space or vector space representations in QSAR modeling. That, to our knowledge, was never addressed in literature. Indeed, this study focuses on 5 datasets - and their functional nature is besides the point as the data set selection criteria was clearly stated in the manuscript – However this reduced test data size masks the fact that several thousands of models were produced just to reach the overall conclusions. The 98 models produced are the result of extensive variable selection procedures, dedicated to identify the best possible models from each (data set–model) pair where, for each one, several hundred models were produced. Just for these five datasets there was a significant computational burden, and we believe they represent fairly large and diverse problems that are very common in many QSAR studies. Furthermore, the test statistics produced (the usage of the Friedman test for model comparison is another novelty in Cheminformatics, we believe) took the size of this reduced set into account and nonetheless being able to produce significance values. Testing hundreds of cases would probably just strengthen our conclusions, as the results appear quite robust.
Round 2
Reviewer 1 Report
None
Reviewer 3 Report
As far as my knowledge of math goes, ANY "metric" space can be embedded into a descriptor space. That is - if you have a METRIC verifying triangle inequalities and the key rules d(A,A)=0, d(A,B)=d(B,A) for any items A and B, then (given a finite set of items) there exists some N-dimensional Euclidean space into which items can be embedded by position vectors x(A), i=1..N, so that d(A,B) represents the precise Euclidean distance ||x(A)-x(B)||. This is called "embedding" or "distance geometry". There are no explicit descriptors on the basis of which you may calculate BLAST scores, but if you take the whole PDB and generate the pairwise distance matrices, then you CAN generate an implicit set of descriptors that perfectly mimick BLAST scores. I'm not talking about approximate solutions with information loss - embedding is 100% rigorous. Could you please reply that if I were to add a NEW protein to the implicit descriptor space I just mentioned, it might NOT fit in - because the final dimensionality of the embedding space is not absolute, but rather dependent on the items it is supposed to embed. Adding a protein with a completely new sequence and rerunning the embedding procedure would recreate a completely different implicit vector space. So the new protein cannot be properly dealt with if I insist to work in a vector space - but would pose no problem in a metric space approach. Except that... a completely new item that is very dissimilar from any training item, to the point of not being embeddable in the implicit descriptor space defined by the trainning set... is by any means a very risky prediction to attempt. Machine learning works by interpolation and some limited extrapolation. So, albeit I do not agree that the approaches are in any way fundamentally different, I agree that metric-based approaches MAY do a better job in terms of extrapolation.
But this is not something you may claim based on FIVE examples of QSAR. You are set out to compare methods in terms of their intrinsic predictive power. So, please, consider the CHEMBL database, extract a few hundreds of QSAR sets, and do a systematic comparison. Your claim that metric-based methods have intrinsic advantages can only be upheld if this happens in GENERAL, not when it happens in 5 cases. So I need a statement like "Out of the 500 QSAR studies, metric-based was better in 29% of cases, fingerprint-based was better in 11% of the cases and the two methods did not significantly differ for the remaining situations". And a honest benchmarking would also request an assessment of computational effort - metric-based is typically more time and memory-consuming, so take care to include both smaller and larger training sets, and not cover only GPCRs and ion channels - also include physico-chemical properties, to make sure you have covered an as large spectrum of possible situations. Then, and only then, you can claim that one class of methods is better than the other . So far, you have shown some advantages based on 5 examples, but I could easily find 5 counterexamples. If you want the community to invest into metric-based methods (which are around for several tens of years by now), maybe you need to prove that the effort is worth it. Report averages and standard deviations of expected benefits, against average+/-stddev of additional cost. Then users will have an objective reason to make a choice. Thanks.